# Wealth-based inequity in full child vaccination coverage: An experience from Mali, Bangladesh, and South Africa. A multilevel poison regression

**Frehiwot Birhanu** [1]*, **Kiddus Yitbarek** [2,3]

**1** School of Public Health, College of Health Science, Mizan-Tepi University, Mizan Aman, Southwest Ethiopia, **2** Department of Health Policy and Management, Institute of Health, Jimma University, Jimma, Ethiopia, **3** School of Public Health, Faculty of Health, University of Technology Sydney, Sydney, Australia

\* frehiwotb2017@gmail.com

**Data Availability Statement:** All files are available on the DHS program dataset (https://www.dhsprogram.com/). In the data source/extraction section, the general website for data in each

## Abstract

### Objectives

Every child around the globe should get routine childhood vaccination, which is mostly affected by the country's economic capacity besides the socioeconomic differences. However, how well countries with different economic capacities address equitable child vaccination remains unanswered.

### Methods

Cross-sectional data from the latest Demographic and Health Survey (DHS) database of Mali, Bangladesh, and South Africa was used for this study. The dependent variable was full child vaccination, and wealth-based inequality was assessed using rate-ratio, concentration curve, and concentration index. A multilevel Poisson regression analysis was used to determine the predictors of inequalities. A risk ratio (RR) with a p-value of 0.05 was used to declare statistical significance. All analysis was weighted.

### Results

Full child vaccination status was 30.15%, 62.18%, and 46.94% in Mali, Bangladesh, and South Africa respectively. Even if the disparity is higher in Mali, the full vaccination favors the better-off family both in Mali, and Bangladesh respectively [CInd: 0.05, 95% CI: 0.01, 0.09], [CInd: 0.02, 95% CI: 0.001, 0.03], wealth status did not have an effect in South Africa. The multilevel poison regression indicated maternal age, occupation, wealth of household, and frequency of watching television to positively affect full vaccination, whereas the number of children in the house negatively affected full vaccination.

### Conclusion

Wealth-based inequality in child vaccination was higher in Mali followed by Bangladesh. There was no observable significant equity gap in South Africa. Wealth status, maternal

countries the program covers are provided as: https://www.dhsprogram.com/data/available-datasets.cfm.

**Funding:** The author(s) received no specific funding for this work.

**Competing interests:** The authors have declared that no competing interests exist.

**Abbreviations:** BCG, Bacille-Calmette-Guérin; CInd, concentration index; EPI, Expanded Immunization Program; GVAP, Global Vaccine Action Plan; MCV, Measles-Containing Vaccine; OPV, Oral Polio Vaccine.

occupation, maternal age, frequency of watching television, and number of children were predictors of full child vaccination.

## Introduction

Despite some differences across countries, a child receives; one dose of the Bacille Calmette-Guérin (BCG) vaccine, three doses of the pentavalent vaccine, one dose of measles-containing vaccine (MCV), three doses of the oral polio vaccine (OPV), and three doses of the pneumo-coccal conjugate vaccine to be declared as fully vaccinated. Mali [1], Bangladesh [2], and South Africa [3] follow the same standard of child vaccination. Although there is a lot of success in eliminating vaccine-preventable diseases in developed countries, children in low-income countries are not still obtaining sufficient services because of the limited health system capacity, problems with the acceptability of the vaccine, and economic barriers to the service as a few reasons among all [4–7].

The complete or partial lack of those vaccines, mostly in low and middle-income countries, contributes to around three million preventable child losses each year [8, 9]. This full vaccination for children could not be achieved in most countries of the world [10]. To make changes in this regards the World Health Organization (WHO) launched the Expanded Immunization Program (EPI) in 1974 [11] Since its introduction the EPI has increased the global uptake for the four core vaccination (BCG, DPT, OPV, and MCV) from 5% in 1973 to over 80% in 2009 [12, 13]. Afterward, in 2012 the World Health Assembly of 194 member states endorsed the Global Vaccine Action Plan (GVAP) intending to put a framework to prevent millions of deaths by 2020 through equitable access to vaccines across the world [14].

The GVAP was a step toward universal health coverage (UHC) [15]. The World Health Organization (WHO) defines UHC as "all people have access to the health services they need, when and where they need them, without financial hardship. It includes the full range of essential health services, from health promotion to prevention, treatment, rehabilitation, and palliative care" [16]. Equity is an inherent dimension, which helps to systematically follow the status of UHC of different health service interventions. Despite these facts, full child vaccination has continued to be pro-rich in many low-income countries [17–19].

Previous studies in this area depict a social gradient in utilizing full child vaccination within a country [20–22]. In most cases, economically better-off countries are better in their health systems to give better health access to their citizens. Many low-income countries, on the other hand, are short of many aspects to fulfill the health service needs of the people [23, 24]. On the contrary, there is evidence indicating that poor people in economically good countries are still falling behind [25, 26]. As per our best search, how far the economically well countries went to fill the service gap between the poor, and rich within the country and how they are different from low-income countries is not well studied.

Besides, variables deemed important to assess the social determinants of health according to the WHO-commissioned social determinants of health inequality framework, [27] were examined in this study. The framework illustrates that socioeconomic stratifications like income, education, occupation, gender, and race/ethnicity determine people's place in a social hierarchy (intermediary variables) like living/working conditions, behavioral, and psychological factors; which finally leads to differences in experiencing exposure and vulnerability to health-compromising conditions [27].

In general, theorists frame inequity in health as unavoidable versus avoidable; individual versus group; absolute versus relative, and so on [28]. Given it is a global concern, investigating

both within and between country inequity assists global and local researchers and decision-makers to identify gaps and learn from experiences. Therefore, in this study, we presented the full child vaccination coverage, within and across countries wealth-based inequalities, and the possible predictors of inequalities by randomly selecting three countries (Mali, Bangladesh, and South Africa) from low-income (LI), lower-middle-income (LMI) and upper-middle-income (UMI) countries.

## Materials and methods

### Study setting, and period

In this study, we presented the experience of Mali, Bangladesh, and South Africa. Mali is a low-income sub-Saharan African (SSA) country located in the northwestern direction of Africa with a 20.25 million population [29]. The report from Mali's 2013 DHS shows 33.9% of full child vaccination [30]. Bangladesh is a lower-middle-income (LMI) country located in the eastern part of Asia with around 161.4 million population. According to a nationwide study, around 83% of children are fully vaccinated for the recommended vaccines in Bangladesh [2]. South Africa is the southernmost country located in Africa with a population of 59.62 million. Economically, the country is now under the upper-middle-income (UMI) category with a GDP of US$ 771 billion. According to a recent survey report of South Africa, the full child vaccination is 59.2% [31].

### Data source/extraction

The data for the three countries were obtained from the Demographic and Health Surveys (DHS), conducted after the Global Vaccine Action Plan (GVAP) had been endorsed by all the WHO member states [32]. Accordingly, the 2018 DHS of Mali, 2017 of Bangladesh, and 2016 data sets of South Africa were available and used for this study.

The DHS are nationally representative household surveys that provide up-to-date data in the areas of population, health, and nutrition [14]. It is often conducted across all WHO regions and at multiple points in time within a country. The sample is usually based on a stratified two-stage cluster design to reach the end households.

### Population and sampling

The target population for the DHS survey is all women aged 15–49 and children under five years of age. A two-stage household-based sampling design is used while dividing the country into enumeration areas (EA), A geographic area containing 131 households, then a complete list of EAs that covers the survey country serves as a sampling frame for the study. A further stage of sampling will be conducted to identify study participants. We considered the child data and took children under the age of five as the study population for this study.

### Variables of the study

The dependent variable for this study was full child vaccination status: a single dose of Bacille Calmette-Guerin (BCG); three doses of diphtheria, tetanus toxoids, and pertussis (DTP); three doses of oral polio vaccine; and one dose of measles-containing vaccine (MCV). Accordingly, if the child received all the listed recommended vaccines, he/she was coded as 1, and 0 otherwise.

To identify the social determinants of full child vaccination as per the WHO's conceptual framework, a set of accessible variables from the DHS database was considered for this study. Accordingly, the sex of the child, age of the mother, number of children in the house, ANC

attendance, maternal education, occupation, wealth quantile, frequency of listening to radio, frequency of watching television, distance from the health facility, place of residence were included in the analysis.

To calculate the household's wealth, the DHS program uses a wealth index that has a composite score of the household's living standard calculated using data on the household's ownership of selected assets, and materials then the final wealth quintile will be obtained after valuation and analysis using the principal component analysis. The households will accordingly fall in either of the poorest (1), poorer (2), middle (3), richer (4), and richest (5) quantiles [33]. The maternal level of education was the highest attainment in school and categorized as no education (1), primary education (2), secondary education (3), and tertiary education (4). To categorize places of residence, the DHS program uses country-specific classification to categorize into rural (0) and urban (1). The sex of the child was categorized into male and female; the mother's age in the five years group and the marital status of the mother were included in the analysis.

## Statistical analysis

The wealth-based equity in using vaccination services for children in this study was measured using rate-ratio, concentration curve, and concentration index (CInd). The rate ratio indicates the degree to which the economically better off utilize health services (child vaccination) as compared to poorer quintiles. It was computed by dividing the proportion of service users in the highest wealth quintile by the proportion in the bottom wealth quintile. Then, Concentration curves were plotted considering the cumulative percentage of the full vaccination (y-axis) against the cumulative percentage of the population, ranked by socio-economic variables (x-axis).

To quantify the wealth-based equity differences, concentration index values were computed with respective standard errors and a 95% confidence interval. The concentration index is twice the area between the concentration curve and the line of equality (the 45-degree line). The concentration index value ranges from -1 to +1. The convention is that the index takes a negative value when the curve lies above the line of equality (the 45-degree line), indicating the disproportionate concentration of the health variable (vaccination) among the poor, and a positive value when it lies below the line of equality [34, 35].

The concentration index for t = 1,..., T groups was computed in a spreadsheet program using the following formula according to Fuller and Lury, 1977 [36].

$$C = (p_1 L_2 - p_2 L_1) + (p_2 L_3 - p_3 L_2) + \ldots + (p_{T-1} L_T - p_T L_{T-1})$$

Where; pt is the cumulative percentage of the sample ranked by economic status in group T, and Lt is the corresponding concentration curve ordinate.

A multilevel Poisson regression analysis; a type of analysis designed to adjust for hierarchal and clustered types of data was used to determine factors driving inequalities in full vaccination among children. All the analyses were adjusted for sampling weights. We specified a 2-level model: at level 1 we adjusted individual factors such as child characteristics, mother's characteristics, and household factors; and at the second level, we adjusted for clustering [S1 Table]. We presented findings with risk ratio (RR) and we used a p-value of 0.05 as a threshold to determine statistical significance. A sub-groping analysis of equity differences by place of residence was also performed. We used STATA version– 14 and Microsoft Spreadsheet to analyze the data.

## Ethical considerations

The data for this study was obtained from the Demographic and Health Survey program. For our purpose, we formally requested and completed the agreement and data usage form before proceeding with the analysis. There was no additional ethical approval sought by the authors.

## Results

### Socio-demographic characteristics

About 5801, 5128, and 1881 respondents from Mali, Bangladesh, and South Africa were included in the analysis. In all of the countries, the full vaccination was distributed proportionally to male and female children. In Mali and South Africa, the highest proportion of vaccinated children were from mothers between the age group of 25 and 29 years. In Bangladesh on the other hand, the highest proportion of 1137 [35.7%] respondents were younger mothers between 20 and 24 years. Almost all 1701 [97.3], and 3,147 [98.7] women in Mali and Bangladesh respectively were married. However, the highest proportion of 486 [55.1%] in South Africa were single mothers. The majority 1199 [68.6%] of the mothers of vaccinated children in Mali were not educated. Whereas, the highest proportion in Bangladesh 1583 [49.7], and South Africa 719 [81.5] have completed secondary education. Regarding residence, about three-quarters of mothers with fully vaccinated children in Mali and Bangladesh live in rural areas. In South Africa, the majority were residents of an urban setting [Table 1].

### Vaccination status

Among the vaccines, the highest proportion of children was vaccinated with BCG in all three countries. On the contrary, the lowest proportion of children (46.1%) was vaccinated with three doses of Oral Polio Vaccine in Mali, (65.3%) were vaccinated with one dose of Measles Containing Vaccine in Bangladesh, and (49.9%) were vaccinated with three doses of

**Table 1. Socio-demographic characteristics of mothers and children with full vaccination status for Mali, Bangladesh, and South Africa.**

| Variables | Category | Mali (n = 5801) F [%] | Bangladesh (n = 5128) F [%] | South Africa (n = 1881) F [%] |
|---|---|---|---|---|
| Sex of the child | Male | 860 [49.2] | 1,639 [51.4] | 458 [51.9] |
|  | Female | 889 [50.8] | 1549 [48.6] | 424 [48.1] |
| Maternal age in 5 years age groups | 15–19 | 144 [8.2] | 482 [15.1] | 63 [7.2] |
|  | 20–24 | 408 [23.3] | 1137 [35.7] | 214 [24.3] |
|  | 25–29 | 461 [26.4] | 850 [26.7] | 270 [30.6] |
|  | 30–34 | 352 [20.1] | 518 [16.3] | 190 [21.6] |
|  | 35–39 | 267 [15.3] | 165 [5.2] | 99 [11.2] |
|  | 40–44 | 96 [5.5] | 31 [1.0] | 39 [4.5] |
|  | 45–49 | 20 [1.2] | 6 [0.2] | 6 [0.7] |
| Marital status | Single | 35 [2.0] | 0 [0] | 486 [55.1] |
|  | Married | 1701 [97.2] | 3,147 [98.7] | 369 [41.8] |
|  | Widowed | 8 [0.5 | 13 [0.4] | 10 [1.1] |
|  | Divorced | 6 [0.3] | 28 [0.9] | 18 [2.0] |
| Level of education | No education | 1199 [68.5] | 187 [5.9] | 8 [1.0] |
|  | Primary | 219 [12.5] | 834 [26.2] | 60 [6.8] |
|  | Secondary | 297 [17.0] | 1583 [49.7] | 719 [81.5] |
|  | Higher | 34 [1.9] | 584 [18.3] | 95 [10.8 |
| Residence | Urban | 408 [23.3] | 843 [26.5] | 529 [59.9] |
|  | Rural | 1341 [76.7 | 2345 [73.6] | 354 [40.1] |
| Wealth quintile | Poorest | 298 [17.0] | 640 [20.1] | 210 [23.8] |
|  | Poorer | 356 [20.4] | 654 [20.5] | 204 [23.1] |
|  | Middle | 415 [23.7] | 600 [18.8] | 197 [22.3] |
|  | Richer | 300 [17.1] | 619 [19.4] | 154 [17.5] |
|  | Richest | 380 [21.8] | 675 [21.2] | 117 [13.3] |

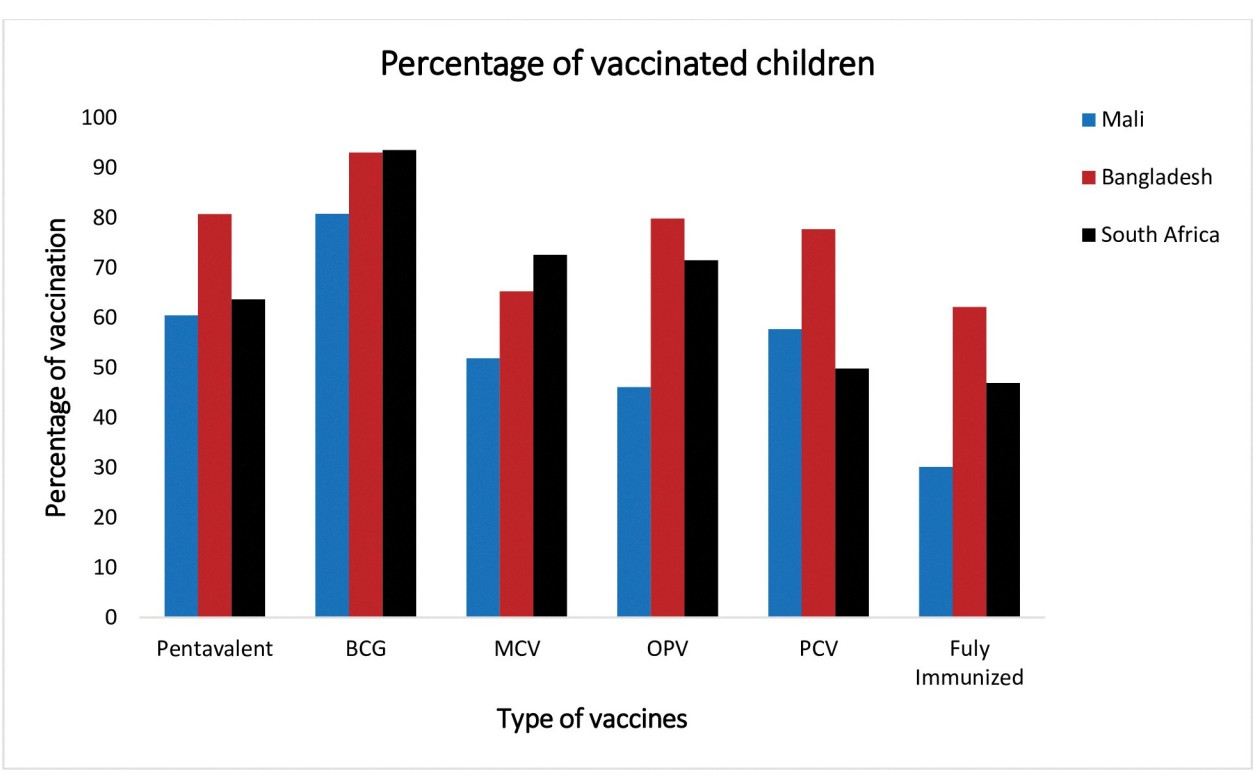

**Fig 1. The percentage of vaccinated children in Mali, Bangladesh, and South Africa in 2018, 2017, and 2016 respectively.**

Pneumococcal conjugate vaccine in South Africa. Overall, 30.2%, 62.9%, and 46.9% of children were fully vaccinated in Mali, Bangladesh, and South Africa respectively [Fig 1].

## Wealth-based inequity in full-vaccination

The concentration curve and index values revealed that in Mali there is a significant disparity in the full vaccination status of children between the lowest and highest wealth households [CInd: 0.05, 95% CI: 0.01, 0.09]. When we compare the lowest wealth, 20%, and the highest 20% in Mali, there is almost a 40% difference in favor of the highest-wealth households. The effect of wealth difference in full vaccination in Bangladesh favors the better-off family. However, in South Africa, full vaccination favors the poor even if the effect is not significant [Fig 2]. The sub-group analysis revealed the equity gap to be severe in rural settings compared to the urban in Mali. Children from wealthy families in rural areas benefit more than the poor. On the contrary, the rural settings children from poor families benefit more in South Africa. [Fig 3]

## The effect of wealth differences on full-child vaccination

Our multilevel poison regression model revealed several predictors of full-child vaccination. Considering other variables constant, the age of the mother was a positive factor affecting full child vaccination in all countries. Similarly, having an occupation was a positive driver in Mali, and Bangladesh while it negatively affected women in South Africa. Additionally, the frequency of watching television was the other positive factor. On the other side, the number of children in the house; richer, and poorest compared to the richest household was negatively associated with full child vaccination [Table 2].

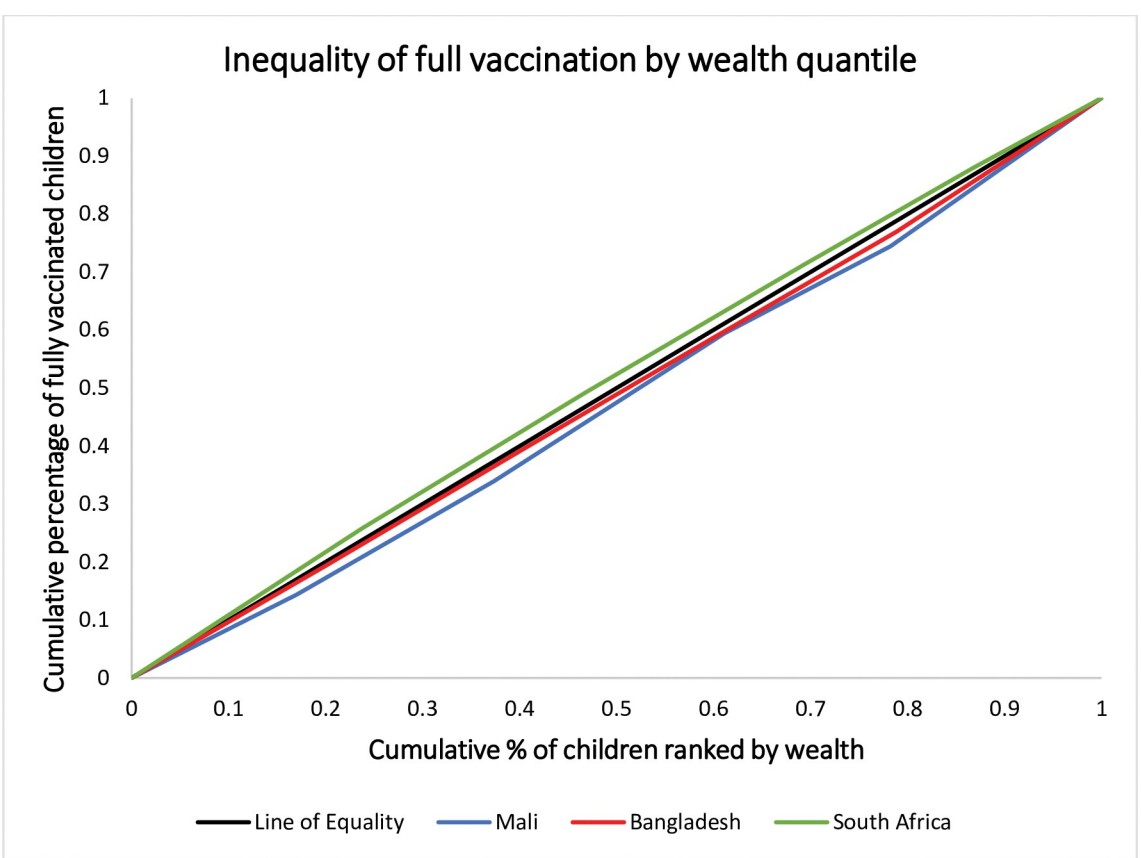

**Fig 2. Concentration curve indicating equity differences in full child vaccination in Mali (2018), Bangladesh (2017), and South Africa (2016).**

## Discussion

For several years, eliminating the socio-economic-driven unnecessary gaps in health and health services use has been the target of various international goals including the Sustainable Development Goals (SDGs) [37]. Despite this, health service use is reported to be pro-rich in many low and middle-income countries [38]. Therefore, this study presented the experience of full child vaccination coverage, within and across country wealth-based inequalities, and predictors by taking Mali, Bangladesh, and South Africa.

The full child vaccination coverage was relatively better (62.9%) in Bangladesh, a LMI country. The result of good child vaccination coverage in Bangladesh might be attributed to the government's initiative to implement a countrywide vaccine campaign like the Measles-rubella campaign in 2014. This increased the vaccination coverage by more than 75% and was certified by WHO as a polio-free country in 2014 [39]. Moreover, Bangladesh's history in child vaccination coverage from the lowest base was exemplary for its contribution to the high reduction of childhood morbidity and mortality and received two GAVI best performance awards in 2009 and 2012 [40].

In our study, the full vaccination coverage in South Africa (46.9%) was lower than in Bangladesh but higher than in Mali (30.2%). In South Africa, while vaccine-preventable diseases are still a threat, a report in 2016 revealed that full child vaccination is deteriorating due to incomplete data, delayed national EPI surveys, inaccurate stock control, and poor cold-chain management [41]. On the other side, though Mali set priority agendas to increase full child

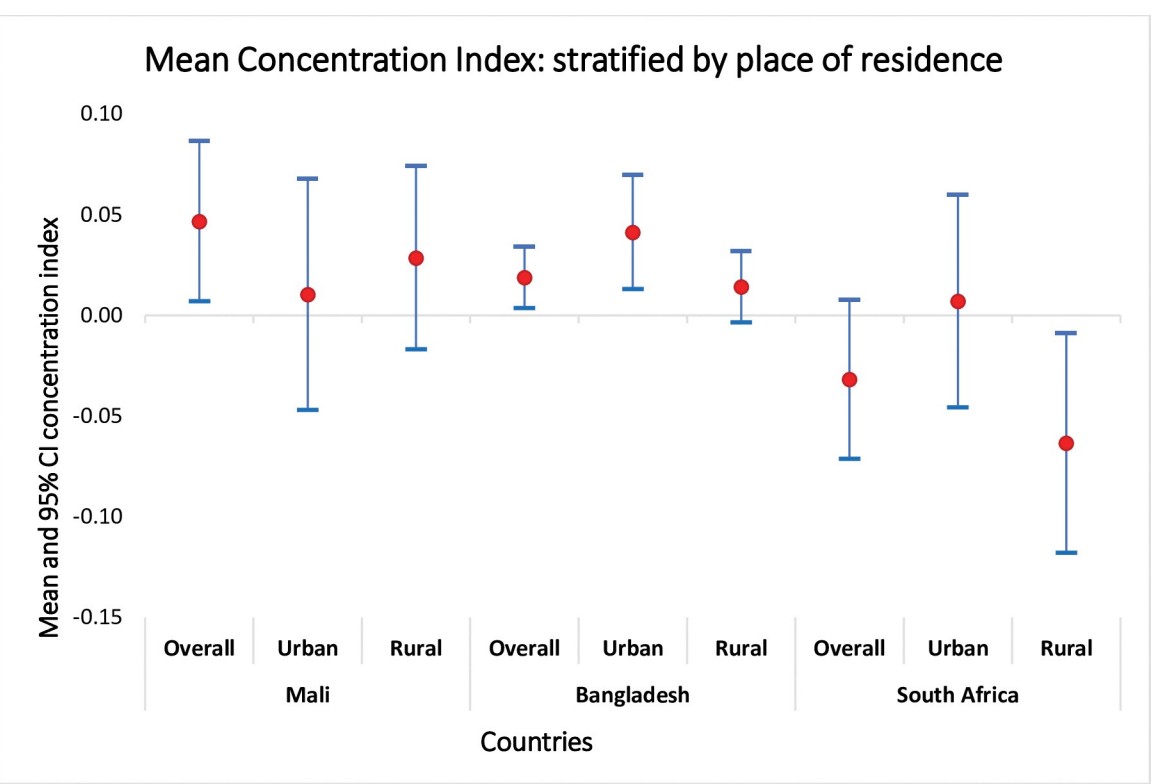

**Fig 3. Concentration index of equity differences stratified by place of residence (urban-rural) in child vaccination in Mali (2018), Bangladesh (2017), and South Africa (2016).**

vaccination by creating community health centers and responsible health committees in 2003; and deploying community health workers in 2011 [42, 43], our findings revealed that; Mali, has the least vaccination coverage.

The other finding in our study revealed that wealth-based Inequality favoring the better-off families was higher in Mali, followed by Bangladesh. In Mali, a 40% difference was observed, whereas it was 22% in Bangladesh. This result is commensurate with the findings of a study in other low-income countries [8, 44, 45]. This indicates that the healthcare system in many low-income countries is sub-optimal to reach the poor segment of the population. Even if promising progress regarding universal access to care has been documented in different low and middle-income countries like Burkina Faso, Botswana, and India [45, 46] the wealth-based inequality in Mali, Bangladesh, and other similar countries poses a great burden towards the global target of ending preventable child mortality by 2030 [47].

We can see the reasons both from the users' and service providers' perspectives. From the user's viewpoint, access to education, access to the media, transportation access, and other many important aspects are low which, affects citizens of the country, especially the poor to use health services [48, 49]. From the service providers' perspective, the low coverage of infrastructure tackles the health system to address people living in remote and slum areas [50]. Disproportionate allocation of the national budget for health services in many low-income countries restrains the health system from reaching those hard to reach economically poor proportion of society [51].

This multilevel poison regression also revealed that the richest proportion of the community in Mali experienced a better vaccination status. Compared to the richest, children from

**Table 2. The relationship between full child vaccination and equity parameters in Mali, Bangladesh, and South Africa.**

| Variables | Category | Mali | Bangladesh | South Africa |
|---|---|---|---|---|
| | | RR [95% CI] | RR [95% CI] | RR [95% CI] |
| Sex of the child | Male | 1 | 1 | 1 |
| | Female | 1.04 [0.94,1.15] | 1.01 [0.95,1.09] | 1 [0.83,1.2] |
| Age of the mother | | 1.03 [1.01,1.04]* | 1.02 [1.01,1.03]* | 1.02 [1.01,1.04]* |
| Number of children | | 0.95 [0.91,0.98]* | 0.93 [0.89,0.97]* | 0.92 [0.84,1.02] |
| ANC | Less than 4 | 1 | 1 | 1 |
| | 4 or more | 1.2 [1.08,1.34] | 1.1 [1.02,1.19] | 1.19 [0.92,1.53] |
| Mother's education | No education | 1 | 1 | 1 |
| | Primary | 1.05 [0.89,1.23] | 1.06 [0.9,1.24] | 0.73 [0.35,1.51] |
| | Secondary | 1.13 [0.96,1.32] | 1.14 [0.97,1.34] | 0.86 [0.43,1.69] |
| | Higher | 1.11 [0.76,1.63] | 1.06 [0.88,1.27] | 0.87 [0.41,1.87] |
| Occupation | Not working | 1 | 1 | 1 |
| | Working | 1.19 [1.07,1.33]* | 1.13 [1.05,1.22]* | 0.76 [0.61,0.96]* |
| Wealth quintile | Poorest | 0.7 [0.54,0.91]* | 0.99 [0.85,1.14] | 1.66 [1.05,2.64] |
| | Poorer | 0.79 [0.61,1.03] | 0.96 [0.84,1.1] | 1.33 [0.87,2.04] |
| | Middle | 0.86 [0.67,1.09] | 0.92 [0.81,1.04] | 1.23 [0.82,1.84] |
| | Richer | 0.74 [0.61,0.91]* | 0.94 [0.83,1.05] | 1.16 [0.77,1.74]* |
| | Richest | 1 | 1 | 1 |
| Frequency of listening to radio | Not at all | | | |
| | Less than once a week | 0.99 [0.85,1.16] | 1.02 [0.85,1.23] | 1 [0.74,1.34] |
| | At least once a week | 0.95 [0.83,1.08] | 1 [0.79,1.27] | 1.14 [0.92,1.43] |
| Frequency of watching television | Not at all | 1 | 1 | 1 |
| | Less than once a week | 1.19 [1.02,1.38]* | 1.03 [0.9,1.18] | 0.87 [0.58,1.29] |
| | At least once a week | 1.21 [1.05,1.39]* | 1.07 [0.98,1.17] | 1.03 [0.78,1.35] |
| Distance to the health facility | Big problem | 1 | 1 | 1 |
| | Not a big problem | 1.04 [0.92,1.17] | 1.03 [0.96,1.11] | 0.96 [0.77,1.2] |
| Place of residence | Urban | 1 | 1 | 1 |
| | Rural | 1.31 [1.07,1.61]* | 1.06 [0.97,1.15] | 0.93 [0.75,1.17] |

*p-value < 0.05

** p-value < 0.01

the richer and poorest families were 26% and 30% less likely to have a full vaccination. The finding is consistent with a finding in Ethiopia, one of the low-income countries, it shows children from a better-off family have a higher likelihood of being fully vaccinated [52]. Another study also shows that children in Mali were vulnerable to vaccine-preventable diseases [53]. The low vaccine coverage in most low-income countries coupled with high socio-economic disparity in full vaccination status contributes to lower achievements in child health indicators.

The other variable identified was the age of the mother. It was a significant positive driver for child vaccination in all countries included in the study. As age increases by one, the odds of vaccinating the child increase by 2–3%. This finding is similar to a study conducted in Afghanistan, a low-income country, which shows the lower the maternal age the lower the probability of child vaccination [54]. The possible reason might be that; women tend to gain lived experience about the importance of child vaccination either from themselves/ relatives or other neighbors through the years [55].

Maternal occupation was the other variable identified in this study. In Mali, and Bangladesh women having an occupation were 19% and 13% more likely to vaccinate their children

respectively. Whereas in South Africa, those having an occupation were 24% less likely to vaccinate their child. The higher odds of child vaccination in women having an occupation are supported in different studies conducted in LMIC [56, 57]. It is well established that besides being employed gives financial stability to cover the transportation and related costs of child vaccination, it provides better autonomy and decision-making power for a woman [58].

The other variable identified was the number of children in the household. Both in Mali and Bangladesh, as the number of the child increased the probability of vaccinating their child went low by 5, and 7% respectively. This study is commensurate with a finding from a study done in Zimbabwe [59]. This may be explained as coupled with other burdens in the household; higher family size might create difficulties to afford expenditures associated with vaccination. The last variable identified was the frequency of watching television. The odds of child vaccination increased by 19–21% for those watching television at least once or above a week. This finding is supported by different studies in low-income countries [60, 61]. As exposure to mass media is found to be a crucial positive driver to increase child vaccination, intervention shall target to increase the information dissemination to favorably impact mothers' choices.

## Strengths and limitations of the study

The identification of wealth-based inequality both within and across the three economically different countries using their recent nationally representative data makes the strength of the study. Besides this, assessing the possible predictors of the resulting inequality using the WHO-commissioned social determinants of health inequality framework provides a summarized way of tackling the social gradients in health. On the other hand, users of this article need to understand a few limitations while reading the findings of this study. In this study, we used data obtained from the DHS Program database, which was used to analyze key health and health-related indicators at the national level. In most cases, the DHS program collects up to five histories for the respondents, which exposes the data to recall bias. Furthermore, we could not find all relevant variables to assess the determinants of social gradients in health. For this reason, this study might have introduced omitted variable bias in the inferential analysis. Future researchers in this area need to assess the wealth-based inequality including more health systems and behavioral factors.

## Conclusion

In this study, compared to Mali and South Africa the vaccination coverage was higher in Bangladesh. Besides this, the vaccination service was far from reaching the poor families in, Mali, and Bangladesh. In South Africa, inequality was not significant. The most important characteristics that create variation in full vaccination status were maternal age, occupation, number of children in the house, frequency of watching television, and wealth of the household. A further study including multiple countries from each economic category would help to better summarize the inequality, as well as to target an intervention at different economic categories.

## Supporting information

**S1 Table. The multilevel poison regression in full child vaccination and equity parameters in Mali, Bangladesh, and South Africa.**
(DOCX)

## Author Contributions

**Conceptualization:** Frehiwot Birhanu, Kiddus Yitbarek.

**Data curation:** Frehiwot Birhanu, Kiddus Yitbarek.

**Formal analysis:** Frehiwot Birhanu, Kiddus Yitbarek.

**Methodology:** Frehiwot Birhanu, Kiddus Yitbarek.

**Software:** Frehiwot Birhanu.

**Supervision:** Kiddus Yitbarek.

**Writing – original draft:** Frehiwot Birhanu.

**Writing – review & editing:** Kiddus Yitbarek.

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
