## [Decision Letter · Decision Letter 0]

7 Jun 2023

PONE-D-22-32266The Socio-economic Inequity in Full Child Vaccination Coverage: An Experience from Mali, Bangladesh, and South Africa.PLOS ONE

Dear Dr. Birhanu,

Thank you for submitting your manuscript to PLOS ONE. After careful consideration, we feel that it has merit but does not fully meet PLOS ONE’s publication criteria as it currently stands. Therefore, we invite you to submit a revised version of the manuscript that addresses the points raised during the review process.

It is recommended to submit the revised version of the manuscript after carefully updating the manuscript as per reviewer's comments and followings.1. The authors considered three countries Bangladesh, Mali, and South Africa for comparing the vaccination coverage. However, these three countries are quite different from the cultural point of view. Thus, it is very important to justify the selection of these countries.2. The authors used the 2018 DHS of Mali, 2014 of Bangladesh, and 2016 data set of South Africa. However, 2014 BDHS is quite old and the latest data (2017/18 BDHS) set is published a few years back. For comparison, it is also important to bear in mind the time period of the study. Therefore, The results should be updated based on recent data.Please ensure that your decision is justified on PLOS ONE’s publication criteria and not, for example, on novelty or perceived impact.

We look forward to receiving your revised manuscript.

Kind regards,

Md. Moyazzem Hossain

Academic Editor

PLOS ONE

Journal Requirements:

Reviewers' comments:

Reviewer's Responses to Questions

**Comments to the Author**

1. Is the manuscript technically sound, and do the data support the conclusions?

Reviewer #1: Yes

Reviewer #2: No

2. Has the statistical analysis been performed appropriately and rigorously? 

Reviewer #1: No

Reviewer #2: No

3. Have the authors made all data underlying the findings in their manuscript fully available?

Reviewer #1: Yes

Reviewer #2: No

4. Is the manuscript presented in an intelligible fashion and written in standard English?

Reviewer #1: No

Reviewer #2: Yes

5. Review Comments to the Author

Reviewer #1: Author are looking at the socio economic inequity in vaccination coverage. However, they need to give clarity on why and how they have selected variables to represents socio-economic. if only going for wealth and education, they should have gone more in-depth analysis. it is quite well established for the variables they selected. Also the why Bangladesh, Mali and South Africa were compared. These countries are quite different from each other. A rationale explaining comparing three countries are required. The discussion might also explained results providing more context to each countries.

Again Age, Gender, education, wealth and place of residents are variables which are quite well researched to impact child health and vaccination coverage. different linkages between them or more in-depth analysis of one of the variable will bring more useful results and will add to what is already known.

South Africa seems to have no wealth inequality (figure 2). but complete immunization is low in south africa.

The context is missing for comparing these countries and inequality results are contradictory.

The writing is generic at times as well for example: " A further study including multiple countries and exploring many reasons for inequality in lower-income countries would help for a better intervention that reduces the equity gap." The sentence is contradictory itself. why author did not do multi country analysis while the data is available from DHS. Arriving at same solution for all country will be difficult.

Reviewer #2: This article addressed a good topic but they did not applied a sophisticated statistical methods. They used logistic regression, already many articles has been published using this method on this topic in Bangladesh and South Africa. In case of Bangladesh they used DHS-2014 old data file while the recent data set is available, for Mali they used 2018 and for south Africa they used 2016 data file, So, the comparison is meaningless.

Some of the Comments for revision:

* Explain the the research objectives mentioning the research gap in Introduction section.

*Used the Same year data set for all countries and then compare it.

*In Method of Analysis Section (line 167) it was mentioned that P-value is considered 0.25 what is logic beyond this (put a reference) while most of the statistical analysis consider 0.05 for variable selection.

* Line 176: All the analyses were adjusted for clusters and sample weights. Check the cluster effects in the data set while the data is collected from different stratum (clusters).

* Use the appropriate statistical methods addressing the clustering effects.

*Explain How the sample weights adjusted.

6. PLOS authors have the option to publish the peer review history of their article (what does this mean?). If published, this will include your full peer review and any attached files.

Reviewer #1: No

Reviewer #2: No

---

## [Author Response · Author response to Decision Letter 0]

17 Jul 2023

Ref: Submission ID PONE-D-22-32266

Title: The Socio-economic Inequity in Full Child Vaccination Coverage: An Experience from Mali, Bangladesh, and South Africa.

Response to editor and reviewers' comments

Dear Editor,

Thank you very much for giving us a chance to revise our manuscript and for your time reviewing our submission. We revised our manuscript based on the comments raised by the editor, and reviewers. We found all of the comments relevant to improve the article; we now prepared three files for submission including the manuscript main document with track changes, the manuscript main document clean copy, and this rebuttal letter. Please find them below. 

Response to editor comments

The authors considered three countries Bangladesh, Mali, and South Africa for comparing the vaccination coverage. However, these three countries are quite different from the cultural point of view. Thus, it is very important to justify the selection of these countries.

Response: Thank you very much for raising such an important concern. We also believe that it needs a thorough discussion. For this study, to see the experience, and take lessons; we randomly selected three countries from different economic categories and see the full vaccination coverage; and within and across countries inequalities in child vaccination. It is correct that the presented countries are different in terms of culture, religion, and economy. However, our primary objective of this study is not to compare the three countries with the above parameters, but to explore the pro-rich inequalities within each as well as the inequality across the three economies, so this study will serve as an experience for bigger studies at Low and Middle-Income Countries. We have justified this in the last paragraph of the background section. 

The authors used the 2018 DHS of Mali, 2014 of Bangladesh, and the 2016 data set of South Africa. However, 2014 BDHS is quite old and the latest data (2017/18 BDHS) set is published a few years back. For comparison, it is also important to bear in mind the period of the study. Therefore, The results should be updated based on recent data. 

Response: Thank you very much, dear Editor. It is an important concern, and we fully accept that. After we received your comments, we run the analysis again to double-check which dataset we used during the analysis for Bangladesh. We made sure that, the dataset we reported was BDHS 2017/18 despite it being mentioned as 2014. It was just a typo. Accordingly, we presented the 2018 DHS of Mali, 2017/18 of Bangladesh, and 2016 data of South Africa in the revised document.

It is possible to check the analysis using the following STATA codes after waiting for cluster differences. 

Waiting 

gen WGT = v005/1000000

For the descriptive statistics please use [iweight=WGT] just after each line.

Define full vaccination before the concentration curve and index analysis: 

clorenz fulvax, rank(v190)

conindex fulvax, rankvar(v190) bounded limits(0 1)

Journal Requirements:

When submitting your revision, we need you to address these additional requirements.Please ensure that your manuscript meets PLOS ONE's style requirements, including those for file naming. The PLOS ONE style templets can be found at

Response: Thank you very much. We have revised the manuscripts and checked for the PLOS ONE requirements accordingly. 

We note that you have stated that you will provide repository information for your data at acceptance. Should your manuscript be accepted for publication, we will hold it until you provide the relevant accession numbers or DOIs necessary to access your data. If you wish to make changes to your Data Availability statement, please describe these changes in your cover letter and we will update your Data Availability statement to reflect the information you provide.

Response: Dear Editor, for this study, as we have used secondary data from the Measure Demographic and Health Surveys (DHS) upon a reasonable request, the data can be obtained from the DHS website (http://www.dhsprogram.com/). We have provided this statement in the methods section, of the data sources/extraction sub-section. 

Please include captions for your Supporting Information files at the end of your manuscript, and update any in-text citations to match accordingly. Please see our Supporting Information guidelines for more information: http://journals.plos.org/plosone/s/supporting-information.

Response: Thank you very much. We have now included the caption for the supplementary information. 

Response to Reviewer #1

Dear reviewer, 

We would like to thank you for your time in reviewing our manuscript and we appreciate all of your effort to make our work better. Those were great inputs to improve this manuscript. Accordingly, we took enough time to revise the whole manuscript based on your suggestions and forwarded the following responses for all of your concerns point-by-point fashion as follows,

1. Author are looking at the socio economic inequity in vaccination coverage. However, they need to give clarity on why and how they have selected variables to represents socio-economic. if only going for wealth and education, they should have gone more in-depth analysis. it is quite well established for the variables they selected. Also the why Bangladesh, Mali and South Africa were compared. These countries are quite different from each other. A rationale explaining comparing three countries are required. The discussion might also explained results providing more context to each countries

Response: Dear reviewer, we fully accept your comment. We indeed aimed to assess the presence of within and across-country inequality in full child vaccination taking some individual characteristics. In this assessment, we used the WHO-commissioned social determinants of health inequality framework (A Conceptual Framework for Action on the Social Determinants of Health (who.int)) to test equity-related determinants of full vaccination. We explicitly mentioned this in the revised version of the manuscript. To reveal the content of the document very well, as our focus is to study the inequality resulting from the wealth status of the household, we modified the title to “Wealth-based inequity in full child vaccination coverage: An experience from Mali, Bangladesh, and South Africa. A multilevel poison regression”

In this study, to see the experience, and take lessons; we randomly selected three countries from different economic categories and see the full vaccination coverage; and within and across countries inequalities in child vaccination. It is correct that the presented countries are different in terms of culture, religion, and economy. However, our primary objective of this study was not to compare the three countries with the above parameters but to explore the pro-rich inequalities within each as well as the inequality across the three economies, so this study will serve as an experience for bigger studies at Low and Middle-Income Countries. We have justified this in the last paragraph of the background section. 

2. Again Age, Gender, education, wealth and place of residents are variables which are quite well researched to impact child health and vaccination coverage. different linkages between them or more in-depth analysis of one of the variable will bring more useful results and will add to what is already known.

Response: Thank you very much, dear reviewer. You are right; several studies reported these variables. In this study, we followed the WHO-commissioned social determinants of health inequality framework (https://www.who.int/publications/i/item/9789241500852). The framework illustrates the socioeconomic stratifications of the individual in the community like income, education, occupation, gender, and race/ethnicity are individual and community-level factors that determine the inequality in health. We did a detailed and sub-group analysis using the variables wealth, maternal education, and place of residence. Furthermore, the variety of analysis methods that we employed makes our study detailed and different. In the final analysis, we have used these variables that can be accessed from the DHS program of the three countries. We have elaborated this in the 5th paragraph of the introduction section, and methods, and variables of the study sub-section. 

3. South Africa seems to have no wealth inequality (figure 2). but complete immunization is low in south Africa.

Response: To clarify this issue, it is correct that in South Africa, an upper middle-income country, the full child vaccination is 46.9%, which is lower than Bangladesh, a lower middle-income country, but higher than Mali, a low-income country. This result is somehow different from our hypothesis that assumes countries of a higher economic category are most likely to have better full child vaccination coverage due to a relatively better health system’s capacity to provide a better health service. This could be due to other factors, which were not included in this study. On the other side, the full child vaccination favors the wealthiest family in both Mali and Bangladesh though it is higher in Mali. This shows that the service in South Africa is not pro-rich or the poor use the services to the same degree as the rich use. 

4. The context is missing for comparing these countries and inequality results are contradictory.

Response: To explain this, evidence suggests that many economically better-off countries are better in their health systems to give better health access to their citizens. Many low-income countries, on the other hand, are short of many aspects to fulfill the health service needs of the people. Therefore, in this study, we wanted to see the experience of countries with different economic classifications. Accordingly, we selected one country from each of low income, lower-middle-income, and upper-middle-income countries using a random selection method. Then wealth-based inequality was assessed to measure the level of disparity between the poor and rich segments of the population in their status of full child vaccination. As per the results, the inequality was scientifically significant in Mali, and Bangladesh, and the child vaccination was in favor of the wealthiest. But we understand and mentioned that the finding is not generalizable to other countries in the respective economic category and it is the experience of the three countries. 

5. The writing is generic at times as well for example: " A further study including multiple countries and exploring many reasons for inequality in lower-income countries would help for a better intervention that reduces the equity gap." The sentence is contradictory itself. why author did not do multi country analysis while the data is available from DHS. Arriving at same solution for all country will be difficult.

Response: Thank you very much for this valuable suggestion. What we have tried to emphasize in our research is to examine the experience/lesson of three randomly selected countries from different economic categories, and see the inequality in full child vaccination status so that this research will serve as a base for further studies. We have modified the statement at the end of the conclusion section of the revised manuscript. 

Response to Reviewer #2

Dear reviewer: We would like to thank you for your time in reviewing our manuscript. We found all of them to be detrimental to the improvement of the paper. Accordingly, we took enough time to revise the whole manuscript based on your suggestions and forwarded the following responses for all of your concerns point-by-point fashion as follows,

This article addressed a good topic but they did not applied a sophisticated statistical methods. They used logistic regression, already many articles has been published using this method on this topic in Bangladesh and South Africa. In case of Bangladesh they used DHS-2014 old data file while the recent data set is available, for Mali they used 2018 and for south Africa they used 2016 data file, So, the comparison is meaningless.

Response: Dear reviewer, you raised critical issues that help us improve our work. Regarding the inferential analysis, you are right binary logistic regression is common in vaccine coverage studies. We had a thorough discussion with the co-author and having a review of the literature, we changed the inferential analysis to a multi-level poison regression adding other relevant independent predictors. 

1. Explain the research objectives mentioning the research gap in Introduction section.

Response: Thank you very much, dear reviewer, we have revised the whole introduction section, and we have added the research gap and objectives of the study at the end of the introduction section in the revised manuscript. 

2. Used the Same year data set for all countries and then compare it.

Response: Thank you very much for this suggestion. I am afraid that DHS Program conducts surveys on similar years across all countries. We chose to consider the most recent datasets of all countries included in the analysis. Accordingly, for the analysis we used 2018, 2017, and 2016 data from Mali, Bangladesh, and South Africa, these are the latest DHS for the three countries, and we believe these are closest enough to conclude.

3. In Method of Analysis Section (line 167) it was mentioned that P-value is considered 0.25 what is logic beyond this (put a reference) while most of the statistical analysis consider 0.05 for variable selection.

Response: Dear reviewer, thank you for this valuable comment. Since we have changed the method of analysis to a multilevel poison regression, we have removed this statement, and for now, we used a p-value of 0.05 to judge the statistical significance of variables. We have stated this in the methods section under the statistical analysis sub-section.

4. Line 176: All the analyses were adjusted for clusters and sample weights. Check the cluster effects in the data set while the data is collected from different stratum (clusters).

Response: The clustering nature of the DHS data might compromise the generalizability of the findings according to the DHS Program. For this reason, we used sampling weights at each stage of analysis. Moreover, we used a multilevel poison regression model to consider different levels of the data collection process. 

5. Use the appropriate statistical methods addressing the clustering effects.

Response: Thank you for this valuable comment dear reviewer. To minimize the bias associated with the clustering nature of the DHS data, we analyzed the final data using a multilevel poison regression. We have started this discussion in the statistical analysis section of the revised manuscript. 

6. Explain How the sample weights adjusted.

Response: Thank you for this valuable comment. As the DHS weighs data to account for the disproportionate sampling, and non-response rate; applying sampling weight is recommended to restore the representativeness of the data. 

The good thing with DHS datasets is there is a calculated sampling weight by experts for further use. For the women and children dataset, the sampling weight is named v005. Sample weights are calculated with six decimals but they are presented without that. Therefore, we divided the variable (v005) by 1 million and generated another variable (WGT) [code: gen WGT = v005/1000000]. We then used this weight variable in all the analyses. For instance, in the descriptive analysis to see the prevalence of pentavalent 3 vaccine we analyzed it as: [tab h53 [iweight=WGT]]. In the inferential statistics on the other hand we used [svyset [pw=WGT], psu(v021) strata(v023)].

---

## [Decision Letter · Decision Letter 1]

21 Aug 2023

PONE-D-22-32266R1Wealth-based inequity in full child vaccination coverage: An experience from Mali, Bangladesh, and South Africa. A multilevel poison regression.PLOS ONE

Dear Dr. Birhanu,

Thank you for submitting your manuscript to PLOS ONE. After careful consideration, we feel that it has merit but does not fully meet PLOS ONE’s publication criteria as it currently stands. Therefore, we invite you to submit a revised version of the manuscript that addresses the points raised during the review process.

1. Improve the Introduction and Discussion sections based on the most recent article published on this topic considering three countries Bangladesh, Mali, and South Africa.

2. Revise the Methods section as per the reviewer’s comment.

We look forward to receiving your revised manuscript.

Kind regards,

Md. Moyazzem Hossain

Academic Editor

PLOS ONE

Journal Requirements:

Reviewers' comments:

Reviewer's Responses to Questions

**Comments to the Author**

1. If the authors have adequately addressed your comments raised in a previous round of review and you feel that this manuscript is now acceptable for publication, you may indicate that here to bypass the “Comments to the Author” section, enter your conflict of interest statement in the “Confidential to Editor” section, and submit your "Accept" recommendation.

Reviewer #2: All comments have been addressed

Reviewer #3: (No Response)

Reviewer #4: All comments have been addressed

2. Is the manuscript technically sound, and do the data support the conclusions?

Reviewer #2: Partly

Reviewer #3: Yes

Reviewer #4: Yes

3. Has the statistical analysis been performed appropriately and rigorously? 

Reviewer #2: Yes

Reviewer #3: Yes

Reviewer #4: Yes

4. Have the authors made all data underlying the findings in their manuscript fully available?

Reviewer #2: Yes

Reviewer #3: Yes

Reviewer #4: Yes

5. Is the manuscript presented in an intelligible fashion and written in standard English?

Reviewer #2: Yes

Reviewer #3: Yes

Reviewer #4: Yes

6. Review Comments to the Author

Reviewer #2: Please define the multilevel-Poisson regression model in methods section.

Check the multicollinearity among the explanatory variables.

Reviewer #3: The study is well-structured and provides important insights into the factors affecting child vaccination in different economic settings. However, there are a few areas where the authors can consider making improvements to enhance the clarity and relevance of their work:

Abstract Clarity and Precision:

The abstract is a crucial section that provides a snapshot of the study's objectives, methods, results, and conclusions. In this case, the abstract seems to be a bit dense and could benefit from being more concise and precise. Focus on highlighting the main findings, methods, and implications of the study.

Data Source Details:

In the "Data source/extraction" section, consider providing more details about the Demographic and Health Survey (DHS) database, such as its purpose, methodology, and coverage. This would help readers understand the source of the data better.

Equity Analysis Visualizations:

The concentration curves and index values are important visualizations to illustrate wealth-based inequities. Consider including these visualizations in the main body of the paper (not just in the reviewer's comments) to help readers better understand the disparities.

Discussion - Global Context:

In the discussion section, consider placing the study's findings in the broader global context. How do the identified disparities and predictors align with trends observed in other low- and middle-income countries? Discuss any potential policy implications that could be generalized to other regions with similar contexts.

Limitations and Future Research:

While you've discussed the strengths of the study, also include a section on limitations. Address any potential biases in the data, assumptions made during analysis, and any limitations of the chosen methodology. Suggest areas for future research, such as exploring other factors that might contribute to vaccination disparities.

Clarity and Flow:

Proofread the manuscript to ensure smooth readability and clarity. Make sure that each section flows logically into the next and that the writing is concise and precise.

Reviewer #4: Nice work. It would be more solid if the considered time period was same for all three selected countries.

7. PLOS authors have the option to publish the peer review history of their article (what does this mean?). If published, this will include your full peer review and any attached files.

Reviewer #2: No

Reviewer #3: **Yes: **Hosna Salmani

Reviewer #4: No

---

## [Author Response · Author response to Decision Letter 1]

3 Oct 2023

Ref: Submission ID PONE-D-22-32266R1

Title: Wealth-based inequity in full child vaccination coverage: An experience from Mali, Bangladesh, and South Africa. A multilevel poison regression.

Response to editor and reviewers' comments

Response to editor comments

Dear Editor,

Thank you very much for taking the time to review our submission, and for allowing us to revise our manuscript. We revised our manuscript based on the comments raised by the editor and the reviewers. We found all of them to be relevant to improve the article; accordingly, we prepared three files for submission including the clean manuscript, the manuscript with track changes, and this rebuttal letter. Please kindly find the point-by-point response below. 

1. Improve the Introduction and Discussion sections based on the most recent article published on this topic considering three countries Bangladesh, Mali, and South Africa.

Response: We considered your suggestion carefully to revise the introduction and discussion sections and updated them with available recent evidence in the area.

2. Revise the Methods section as per the reviewer’s comment.

Response: Thank you very much. We have now modified the methods section according to the reviewer’s comment. 

Response to reviewer two

Dear reviewer, 

Thank you very much for spending your precious time revising our manuscript and providing your valuable comments. The suggestions you made are important to improve our work. Accordingly, we have used all your comments and made changes to the revised copy of the manuscript. 

1. Please, define the multilevel-Poison regression model in the methods section. Check the Multi-collinearity among the explanatory variables.

Response: Thank you very much again. We described multilevel-poison regression in the last paragraph of the statistical analysis section. Regarding the multi-collinearity, we have checked for the possible correlation between the predictor variables during the data analysis, and no correlations were detected among them. 

Response to reviewer three

Dear reviewer, 

Thank you very much for taking your valuable time to review our manuscript. We appreciate all your efforts to the improvement of our article. All the comments and suggestions you made were very relevant, and we accepted all of them and revised the manuscript's main document accordingly. Below, we explained all of your concerns in a point-by-point fashion as follows:

The study is well structured and provides important insights into the factors affecting child vaccination in different economic settings. However, there are a few areas where the authors can consider making improvements to enhance the clarity and relevance of their work.

1. Abstract clarity and precision: The abstract is a crucial section that provides a snapshot of the study’s objectives, methods, results, and conclusions. In this case, the abstract seems to be a bit dense and could benefit from being more concise and precise. Focus on highlighting the main findings, methods, and implications of the study.

Response: We agree with your comment, and we have modified the abstract section making it more precise and focused accordingly.

2. Data Source Details: in the data source/extraction” section, consider providing more details about the Demographic and Health Survey (DHS) database, such as its purpose, methodology, and coverage. This would help readers understand the source of the data better.

Response: thank you very much again. We have now included the purpose, methodology, and coverage of the DHS program in the data source/extraction section of the revised manuscript. 

3. Equity Analysis Visualizations: The concentration curves and index values are important visualizations to illustrate wealth-based inequities. Consider including these visualizations in the main body of the paper (not just in the reviewer's comments) to help readers better understand the disparities.

Response: Including the illustrations/figures will indeed help the readers a lot to capture the information provided. We have included the concentration curve indicating the equity differences in full child vaccination; and an illustration of the concentration index of equity differences stratified by place of residence (urban-rural) in Figure 2, and 3 respectively, in wealth wealth-based inequality sub-section of the result section of the main manuscript. 

4. Discussion: In the discussion section, consider placing the study's findings in the broader global context. How do the identified disparities and predictors align with trends observed in other low- and middle-income countries? Discuss any potential policy implications that could be generalized to other regions with similar contexts.

Response: We accepted your comment fully. We have now modified it by narrating the scenario with more other LMIC countries, and we have suggested the possible implication for action in paragraph four of the discussion section.

5. Limitations and Future Research: While you’ve discussed the strength of the study, also include a section on limitations. Address any potential biases in the data, assumptions made during analysis, and any limitations of the chosen methodology. Suggest areas for future research, such as exploring other factors that might contribute to vaccination disparities.

Response: We appreciate the suggestion. In the revised copy of our manuscript, we have discussed the possible limitations of our study: the secondary nature of the data does not allow recruiting many variables of interest, recall bias might have been introduced and the possibilities of unseen variable bias also suggested areas for future researchers in the strength and limitation section of the revised manuscript. 

6. Clarity and Flow: Proofread the manuscript to ensure smooth readability and clarity. Make sure that each section flows logically into the next and that the writing is concise and precise. 

Response: Dear reviewer, thanks for your suggestion. We have proofread the whole manuscript and made necessary edits throughout the document.

---

## [Decision Letter · Decision Letter 2]

16 Oct 2023

Wealth-based inequity in full child vaccination coverage: An experience from Mali, Bangladesh, and South Africa. A multilevel poison regression.

PONE-D-22-32266R2

Dear Dr. Birhanu,

We’re pleased to inform you that your manuscript has been judged scientifically suitable for publication and will be formally accepted for publication once it meets all outstanding technical requirements.

Kind regards,

Md. Moyazzem Hossain

Academic Editor

PLOS ONE

Additional Editor Comments (optional):

Reviewers' comments:

Reviewer's Responses to Questions

**Comments to the Author**

1. If the authors have adequately addressed your comments raised in a previous round of review and you feel that this manuscript is now acceptable for publication, you may indicate that here to bypass the “Comments to the Author” section, enter your conflict of interest statement in the “Confidential to Editor” section, and submit your "Accept" recommendation.

Reviewer #2: All comments have been addressed

Reviewer #3: All comments have been addressed

2. Is the manuscript technically sound, and do the data support the conclusions?

Reviewer #2: Yes

Reviewer #3: Yes

3. Has the statistical analysis been performed appropriately and rigorously? 

Reviewer #2: Yes

Reviewer #3: (No Response)

4. Have the authors made all data underlying the findings in their manuscript fully available?

Reviewer #2: Yes

Reviewer #3: Yes

5. Is the manuscript presented in an intelligible fashion and written in standard English?

Reviewer #2: Yes

Reviewer #3: Yes

6. Review Comments to the Author

Reviewer #2: (No Response)

Reviewer #3: Hello,

I wanted to extend my heartfelt gratitude to you for taking my comments into consideration while reviewing the manuscript. It's great to see that the manuscript can now proceed toward publication after addressing the feedback.

Wishing you all the best with the upcoming stages of this project.

Sincerely,

Hosna

7. PLOS authors have the option to publish the peer review history of their article (what does this mean?). If published, this will include your full peer review and any attached files.

Reviewer #2: No

Reviewer #3: **Yes: **Hosna Salmani

---

## [Editor Report · Acceptance letter]

12 Dec 2023

PONE-D-22-32266R2 

Wealth-based inequity in full child vaccination coverage: An experience from Mali, Bangladesh, and South Africa. A multilevel poison regression. 

Dear Dr. Birhanu:

I'm pleased to inform you that your manuscript has been deemed suitable for publication in PLOS ONE. Congratulations! Your manuscript is now with our production department. 

Kind regards, 

on behalf of

Prof. Md. Moyazzem Hossain 

Academic Editor

PLOS ONE